# How We Manage Bone Marrow Edema—An Interdisciplinary Approach

**DOI:** 10.3390/jcm9020551

**Published:** 2020-02-18

**Authors:** Sebastian F. Baumbach, Vanessa Pfahler, Susanne Bechtold-Dalla Pozza, Isa Feist-Pagenstert, Julian Fürmetz, Andrea Baur-Melnyk, Ulla C. Stumpf, Maximilian M. Saller, Andreas Straube, Ralf Schmidmaier, Jan Leipe

**Affiliations:** 1Comprehensive Osteology Center Munich, University Hospital, Ludwig-Maximilians-University Munich, 80336 Munich, Germany; Sebastian.Baumbach@med.uni-muenchen.de (S.F.B.); Vanessa.Pfahler@med.uni-muenchen.de (V.P.); Susanne.Bechtold@med.uni-muenchen.de (S.B.-D.P.); Isa.Pagenstert@med.uni-muenchen.de (I.F.-P.); Julian.Fuermetz@med.uni-muenchen.de (J.F.); Andrea.Baur@med.uni-muenchen.de (A.B.-M.); ulla.stumpf@med.uni-muenchen.de (U.C.S.); Maximilian.Saller@med.uni-muenchen.de (M.M.S.); Andreas.Straube@med.uni-muenchen.de (A.S.); 2Department of General, Trauma and Reconstructive Surgery, University Hospital, Ludwig-Maximilians-University Munich, Nussbaumstraße 20, 80336 Munich, Germany; 3Department of Radiology, University Hospital, Ludwig-Maximilians-University Munich, Marchioninistraße 15, 81377 Munich, Germany; 4Department of Pediatric Endocrinology and Diabetology, University Hospital, Ludwig-Maximilians-University Munich, Lindwurmstraße 4, 80337 Munich, Germany; 5Department of Orthopaedic Surgery, Physical Medicine and Rehabilitation, University Hospital, Ludwig-Maximilians-University Munich, Marchioninistraße 15, 81377 Munich, Germany; 6Experimental Surgery and Regenerative Medicine (ExperiMed), Department of General, Trauma and Reconstructive Surgery, Ludwig-Maximilians-University (LMU), Fraunhoferstraße 20, 82152 Planegg-Martinsried, Germany; 7Department of Neurology, University Hospital, Ludwig-Maximilians-University Munich, Marchioninistraße 15, 81377 Munich, Germany; 8Department for Endocrinology and Diabetology, Department of Internal Medicine IV, Ludwig-Maximilians-University Munich, Ziemssenstraße 1, 80336 Munich, Germany; 9Division of Rheumatology and Clinical Immunology, Department of Internal Medicine IV, Ludwig-Maximilians-University Munich, Ziemssenstraße 1, 80336 Munich, Germany; 10Division of Rheumatology, Department of Medicine V, University Hospital Mannheim, Medical Faculty Mannheim of the University Heidelberg, Ludolf-Krehl-Straße 13–17, 68167 Mannheim, Germany

**Keywords:** Bone marrow edema (BME), magnetic resonance imaging (MRI), management, diagnosis, treatment

## Abstract

Bone marrow edema (BME) is a descriptive term for a common finding in magnetic resonance imaging (MRI). Although pain is the major symptom, BME differs in terms of its causal mechanisms, underlying disease, as well as treatment and prognosis. This complexity together with the lack of evidence-based guidelines, frequently makes the identification of underlying conditions and its management a major challenge. Unnecessary multiple consultations and delays in diagnosis as well as therapy indicate a need for interdisciplinary clinical recommendations. Therefore, an interdisciplinary task force was set up within our large osteology center consisting of specialists from internal medicine, endocrinology/diabetology, hematology/oncology, orthopedics, pediatrics, physical medicine, radiology, rheumatology, and trauma surgery to develop a consenus paper. After review of literature, review of practical experiences (expert opinion), and determination of consensus findings, an overview and an algorithm were developed with concise summaries of relevant aspects of the respective underlying disease including diagnostic measures, clinical features, differential diagnosis and treatment of BME. Together, our single-center consensus review on the management of BME may help improve the quality of care for these patients.

## 1. Introduction

Bone marrow edema (BME) is a descriptive term for a common finding in magnetic resonance imaging (MRI). It can occur in virtually all bones, but is most often observed in the lower extremities [1,2,3]. BME can be symptomatic or asymptomatic. Therefore, painful BME needs to be differentiated from incidental BME. This manuscript covers the diagnosis and treatment of painful BME.

Painful BME can occur spontaneously (primary BME: BME syndrome (BMES)) or secondary to various diseases covering almost all medical specialties (secondary BME). Consequently, the differentiation between primary and secondary BME, as well as the identification of the underlying condition in secondary BME, is of utmost importance to initiate the proper treatment. However, interdisciplinary, evidence-based guidelines for the diagnosis and management of BME are lacking. This is problematic, since patients suffering painful BME are seen by various different medical professions, independent of whether their clinical background covers the underlying pathology. Additionally, different medical specialties often follow their own diagnostic path. In the absence of interdisciplinary management algorithms, this frequently leads to a repetition of diagnostic measures (resulting in additional heath care costs), an unsatisfactory patient journey, missed/incorrect or delayed diagnosis, and undertreatment with potentially deleterious consequences.

The lack of an interdisciplinary, evidence-based diagnostic algorithm led us to initiate a project for our level 1 university hospital, aiming to develop consensus diagnostic algorithms based on the available evidence and expert opinion. An interdisciplinary task force was set up at Ludwig-Maximilians-University (LMU) in Munich consisting of specialists from internal medicine, endocrinology/diabetology, hematology/oncology, orthopedics, pediatrics, physical medicine, radiology, rheumatology, and trauma surgery; these specialists reviewed the literature in their field, collected practical experiences, and developed an algorithm over nine months. This algorithm was presented at different national meetings, critically discussed and adapted. The results were redefined, summarized, and finally used as as a standard operating procedure at our university hospital.

In this interdisciplinary manuscript, the LMU Consensus Group presents our algorithm for the assessment and management of BME, along with a comprehensive overview of different pathologies/diseases potentially underlying BME. This is not a guideline or recommendation of a national medical society, but rather a single-center perspective with recommendations, derived from review of the literature and practical experience, that may be critically discussed and adapted to other facilities. Nevertheless, it is to our knowledge the first published perspective on the interdisciplinary management of BME. It may help to improve the quality of care for these patients until guidelines are published with higher grade evidence.

## 2. Histopathology and Molecular Mechanisms of BME

Although the altered BME signal pattern observed in MR images is probably related to a displacement of normal fatty bone marrow by a more water-rich material or increased tissue vascularity [4], the actual histopathological mechanism of BME remains unknown [5]. To address this point, we conducted a thorough literature review to identify studies that investigated the underlying molecular, immunological and histopathological findings in BME of various causes. These studies are summarized and grouped per their appearance in the manuscript in Appendix A.

As a summary, studies investigating inflammatory causes for BME, such as spondyloarthropathy, ankylosing spondylitis or rheumatoid arthritis, show an increased vascularity and cellularity, mainly consisting of cells of the immune system (T cells, B cells, macrophages). In contrast, studies on BME in advanced osteoarthritis showed a reduced perfusion, thickened subchondral plate and increased bone resorption. Osteonecrosis somewhat mirrors the pathological changes that are observed in osteoarthritis. However, these histological and molecular changes are not bound to the subchondral bone. Studies on BMES are overall inconclusive and, based on the herein defined criteria, do not always meet the level of diagnostic accuracy required to diagnose BMES.

Overall, the data for any entity of BME is sparce and predominantly limited to situations in which BME already is present. Further analyses of underlying molecular mechanisms are necessary to provide new pathophysiological insights that might lead to novel, disease-specific therapeutic targets.

## 3. Imaging Modalities

This section gives an overview of different imaging modalities frequently applied in the case of painful BME. Their significance within the diagnostic work-up will be further outlined throughout the manuscript and the proposed diagnostic algorithm.

### 3.1. Magnetic Resonance Imaging (MRI)

MRI is an indispensable tool to detect or rule-out various differential diagnoses for BME [6,7]. The sequence protocol for BME should include fat-saturated sequences like STIR or PDW FS in three spatial orientations, as well as an unenhanced T1w-sequence and a fat-saturated T1w-sequence after gadolinium application [8] In the fat-saturated STIR or PDW images, edema of bone marrow and soft tissue, as well as effusions, can be detected [9,10] After application of gadolinium, BME shows an early and strong enhancement [11] whereas necrotic tissues typically show no contrast enhancement [12,13,14,15]. A more sensitive approach to differentiating BME from osteonecrosis is high temporal perfusion imaging with quantitative determination of plasma flow and mean transit time [16].

### 3.2. Computed Tomography (CT)

In case of a suspected leading bony pathology, CT can be helpfull to supplement MRI imaging. CT accurately shows trabecular, cortical or subcortical fractures and thereby contributes to the differential diagnosis of BME, such as subchondral infraction in osteonecrosis, stress fractures, mechanical joint diseases such as osteochondral lesions (OCL) and tumors like osteoid osteoma.

### 3.3. Dual-Energy X-ray Absorptiometry (DXA)

DXA aims at diagnosing osteoporosis in the sense of a systemic skeletal disease with a susceptibility to fracture [17], to monitor a possible progression, and to evaluate response to therapies such as antiresorptive treatment. However, the interpretation of DXA is difficult and its sensitivity is limited [17] In particular cases, the determination of real three-dimensional bone mineral density by quantitative computed tomography (Q-CT) may be helpful for clinical judgement.

### 3.4. Other Modalities

Conventional radiographs today play a minor role in the diagnostics of BME [18,19] but may give a first hint to mechanical causes, such as osteoarthritis, OCL, or advanced avascular necrosis (AVN). Skeletal scintigraphy can detect increased bone turnover and accumulated vascularization [20], but cannot definitively diagnose BME because of its low sensitivity and specificity [7,18,21]. Both imaging modalities might be helpful in selected cases but are not considered an inherent part of the standard diagnostic algorithm. Recently published data suggest that ultrasound may be helpful in the diagnosis of BMES due to its reliable assessment of joint effusion and capsular thickening in patients with BMES of the hip [22].

## 4. Classification

As outlined above, painful BME can originate from various pathologies. In order to help the physician to incorporate a mind-map summary of essential causes of BME, the Ludwig-Maximilians-University (LMU) consensus group has tried to group these causes according to their etiology. Originating from orthopedic literature, it has been recommended to subgroup BME as traumatic [6], mechanical/degenerative [23], or idiopathic/ischemic/metabolic [18,24,25]. The LMU Consensus Group considers these subclassifications to be incomplete and proposes the classification scheme outlined in Table 1.

## 5. Diagnostic Steps

The aim of the interdisciplinary LMU Consensus Group was to provide a stepwise diagnostic workflow guiding physicians towards the proper diagnosis. This will help to reduce the number of consultations, diminish unnecessary diagnostics and lead to an earlier initiation of the correct treatment, thereby promoting a quicker recovery for the patient. The herein algorithm presented in Figure 1 starts with a patient presenting with persisting pain and MRI imaging revealing a BME.

### 5.1. Medical History and Clinical Examination

A thorough medical history should be obtained according to potential underlying causes (Table 1). Important medical conditions to query include (previous) glucocorticoid treatment, clinical signs of Cushing’s syndrome, smoking, alcohol abuse, coagulopathies, weight loss, night sweats, history of infection, status post arthrocentesis. In case of an acute trauma or recent surgical intervention, the patient should be referred to an orthopedic surgeon (Figure 1).

Clinical examination should include vital signs, signs of local infection (erythema, warmth, swelling, pain) and joint effusion.

### 5.2. Basic Laboratory Work-Up

Any patient suffering from non-traumatic, painful BME should undergo a basic laboratory work-up. We recommend including at least differential blood count and inflammatory markers (C-reactive protein (CRP) and erythrocyte sedimentation rate). Elevated inflammation markers could suggest inflammatory or septic disease (Figure 1).

### 5.3. Joint Effusion

Next to traumatic BME, septic arthritis and osteomyelitis (see Section 6.2) are severe diseases necessitating immediate treatment. Initially they can present with subtle symptoms including painful BME, joint effusion and only minimally/moderately elevated CRP. Consequently, in any patient presenting with this combination of symptoms, arthrocentesis and synovial analysis should be performed. Synovial fluid analysis should include nucleated white cell count with differential, polarized light microscopy (to test for crystals), Gram staining, and bacterial culture. White blood cell (WBC) counts ≥2000/mm^3^ are characteristic for inflammatory arthritis and the patient should be referred to a rheumatologist (Figure 1). WBC counts >50,000/mm^3^ are highly suspicious for septic arthritis. These patients need immediate referral to an orthopedic surgeon. However, since lower WBC counts can be observed in early phases of septic arthritis, repeated arthrocentesis may be necessary.

### 5.4. Further Work-Up (Radiographic and Laboratory)

In patients with normal inflammatory markers, further work-up including CT imaging is recommended (Figure 1). The combined interpretation of the MRI and CT images should be conducted by a radiologist experienced in musculoskeletal imaging. At this stage, mechanical/degenerative causes (degeneration/instability, osteochondrosis dissecans, osteoarthritis, stress fractures), AVN, (Charcot) neuro-osteoarthropathy, and neoplasia can be diagnosed. In case of fracture, (Charcot) neuro-osteoarthropathy, or lesions at risk of fracture, the affected extremity should be immobilized and the patient advised to conduct non-weightbearing activities.. Patients should then be referred to an orthopedic surgeon or oncologist, respectively.

If diagnostics did not reveal a cause for painful BME at this stage, we recommend an extended laboratory work-up (Figure 1), as well as a DXA or Q-CT. The combined interpretation of these diagnostic criteria aims at identifying primary and secondary causes of osteoporosis. In case the cause remains unknown, the painful BME should then be classified as BMES.

## 6. Secondary Causes for BME

Secondary causes of BME must not be overseen as they allow for treatment of the underlying pathology causing the painful BME. The different pathologies leading to BME are grouped according to the proposed etiological mind-map (Figure 1). The diagnostic approach comprises a thorough history and clinical examination and extended laboratory workup (see within the flow chart) as outlined below. The main categories of secondary causes are traumatic, septic, primary inflammatory/rheumatic, mechanical/degenerative, neoplastic, ischemic and metabolic.

### 6.1. Traumatic

Traumatic BME directly or indirectly resulting from trauma include traumatic BME or (micro-) fracture, post-surgical BME, and complex regional pain syndrome. Depending on the force, load transmission and bone quality, a trauma initially causes a lesion/fracture of the trabecular bone microarchitecture (traumatic BME) that can progress to cortical fractures (typical fracture) and fracture dislocation [26]. Trabecular disruption causes the increased fluid levels detected by MRI. The affected region (including both adjacent joints) should be immobilized and the patient advised to not bear any weight on the affected extremity. Whereas isolated traumatic BMEs mostly resolve within 2–4 months following conservative treatment [27], fractures necessitate specific treatment consisting of a wide variety of of conservative and surgical. Post-surgical BME might persist for more than one year [28,29,30,31]. The pathophysiology probably involves direct intraoperative trauma/ischemia and/or altered biomechanics [32]. The patient should be referred to the treating surgeon for re-evaluation. If infectious causes or revision surgery are ruled out, pharmacological treatment has been applied successfully [29]. Complex regional pain syndrome (CRPS) can develop secondary to any trauma [33]. The autonomic nervous system seems to be important in the acute phase triggering a neurogenic inflammation with release of pro-inflammatory cytokines [33,34]. This inflammation subsequently triggers nociceptive fibers to release neuropeptides, which cause vasodilatation and extravasation of proteins. Clinically, this stage is characterized by edema and hyperemia. At later stages the affected region often becomes dystrophic and colder than the contralateral site [35]. CPRS is diagnosed by the clinical presentation of continuing pain that is disproportionate to the triggering trauma, mostly based on Budapest criteria [36]. Typical MRI findings are patchy BME and diffusely increased juxta-articular fluid. The treatment of the CRPS consists of two approaches: (1) a sensory reconditioning by careful physiotherapy (including mirror-therapy) and (2) a pharmacological therapy with cortisone (early stages), bisphosphonate, gabapentin, ketamine and topical dimethylsulfoxide (DMSO). In chronic, therapeutically refractory conditions, patients may benefit from spinal cord stimulation or intrathecal baclofen [33].

### 6.2. Septic BME

Red-flags for septic BME (osteomyelitis/osteitis, septic arthritis) are age >80 years, implants, recent joint injection or surgery, intravenous drug abuse, skin lesions, and immune defects including diabetes in addition to typical clinical and biochemical findings. (Figure 1). Whenever suspected, arthrocentesis or bone biopsies should be performed.

### 6.3. Primary Inflammatory/Rheumatic BME

In rheumatology, BME is of great importance for the diagnosis of musculoskeletal inflammation including arthritis, spondylitis and enthesitis. Subchondral BME lesions in arthritis result from osteitis characterized by infiltrates of lymphocytes, plasma cells and macrophages that are related to osteoclasts replacing the bone marrow fat BME can be observed already after a few weeks after the onset of symptoms [37] and becomes attenuated after effective anti-inflammatory treatment [38]. Subchondral BME is a strong predictor of subsequent joint destruction [34,35,36]. The current guidelines recommend early therapy with glucocorticoids and disease-modifying anti-rheumatic drugs, particularly methotrexate [39].

In axial spondyloarthritis (axSpA), BME at spine and sacroiliac joints is considered an early sign of axial inflammation and associated with histological inflammation, clinical symptoms and radiographic progression [40,41]. TNF and IL-17 are important cytokines driving this process [37]. Its characteristic features of axSpA are age <45, inflammatory back pain, HLA-B27 and extra-articular manifestations. The first-line therapy is NSAIDs. If two different NSAIDs are insufficient to control the inflammation, biologic agents (inhibitors of TNF and IL-17) are recommended. *Enthesitis* is most common in psoriatic arthritis (PsA) and axSpA [42]. Histopathologic studies revealed lymphocytic infiltrates reflecting local inflammation and new bone formation (enthesophytes), with the cytokines TNF, IL-17 and IL23 being pathophysiologically important [37]. The first- line treatment consists of NSAIDs. In case of NSAID failure, e.g., for PsA, biologic agents (inhibitors of TNF, IL-17 and IL-12/23) and the PDE-4 inhibitor apremilast are recommended [43]. Chronic non-bacterial osteomyelitis (CNO) is a rare inflammatory disease affecting children and adults. It occurs at the metaphyses of long bones, pelvis, vertebral column, shoulders/clavicles and mandibles [38]. CNO is often associated with other inflammatory diseases, such as psoriasis, palmoplantar pustulosis and inflammatory bowel disease. The primary clinical sign is pain, but additional symptoms may be caused by paraosseous inflammation, including swelling, warmth and erythema [39]. The treatment includes NSAIDs, corticosteroids, bisphosphonates, sulfasalazine, methotrexate and biologicals such as TNF inhibitors [39].

### 6.4. Mechanical/Degenerative

Mechanical/degenerative BME includes osteoarthriris, insertional tendinopathies, (osteo)chondral lesions and bone stress injuries.Osteoarthritis (OA) is no longer believed to be a disease of cartilage degeneration, but rather a combined pathology involving the synovium and subchondral bone [5]. MRI and histopathological studies found that BME in osteoarthritic bone resembles a combination of fibrosis and bone marrow necrosis more than it resembles edema [5,44] and should therefore be termed bone marrow lesions (BML) instead of BME [5]. BML have been correlated to pain and are predictive for joint replacement [40,41,44,45,46,47]. BML might be a possible target for future OA treatment strategies. Several RCTs were able to show that bisphosphonate therapy resulted in a considerable reduction in BML size [48] and pain [42] in patients suffering painful OA with BML. Valid data showing a positive effect on the progression of OA are still missing.

Insertional tendinopathy is attributed to a combination of mechanical and biochemical causes affecting most often the fibrocartilaginous tendon insertional of the rotator cuff, extensor carpi radialis brevis, ligamentum patellae and the Achilles tendon. It hardly ever demonstrates histological evidence of inflammation but rather degenerative changes [49]. Despite limited evidence, non-surgical treatments including eccentric exercises, shock wave therapy, and injections of platelet rich plasmas are sometimes used as therapy [50,51,52]. Glucocorticoid injections should be used with caution given the risk of tendon degeneration. Bisphosphonates, which have been shown to reduce pain, can be considered off label [43]. Surgical treatment is only indicated in case of failed and extensive conservative treatment. Chondral or osteochondral lesions can be asymptomatic or symptomatic [53] and occur in any synovial joint, predominantly the knee, ankle, talus, hip, shoulder and elbow [54,55,56,57,58,59]. Various etiological theories have been proposed, including family history, local ischemia, rheumatic disease, acute or repetitive trauma [60]. Any first episode of a symptomatic lesion without acute trauma should be treated conservatively, including rest, immobilization, NSAIDs and restriction of activity. In case of persistent pain, various treatment strategies have been proposed, with osteochondral autograft transfer system (OATS) and autologous matrix-induced chondrogenesis (AMIC) representing the most promising [59,61,62]. Bone stress injuries occur after repetitive high forces or atypical forces due to joint instability. Microfractures present as BME on MRI and their accumulation can result in stress fractures [63]. Although bone stress injuries can occur in any bone, the lower leg (40%) and foot (35%), are most commonly involved [64,65]. The key to successful treatment of bone stress injuries is early diagnosis [66] and identification of the nature of the injury (stress lesion (reaction), stress fracture, or instability).

### 6.5. Neoplastic

Although quite rare, neoplastic causes should be kept in mind when evaluating BME. *Solid tumours* may induce reactive marrow edema. Primary cancer of the bone is rare. The most frequent are osteosarcoma, chrondrosarcoma and Ewing sarcoma which usually occur in children. Metastases of the bone are more frequent with 90% of all skeletal metastases represented by myeloma-, breast-, prostate- or renal cancer. Almost the only painful benign bone lesion that can mimic BME is osteoid osteoma, occurring in children and young adults. Hematological neoplasms may lead to suspicious MRI findings, including BME. Acute leukemias and chronic myeloproliferative neoplasia (MPN) can be diagnosed by blood cell workup. Pathological fractures and osteolyses are typical manifestations of multiple myeloma (MM), which similarly involves the bone marrow of the body. It can be diagnosed by the M spike in serum protein electrophoresis; immunofixation proves clonality. All MM patients need cytotoxic treatment and high-dose bisphosphonates. Indolent systemic mastocytosis (ISM) is a hematological disease that may cause osteoporosis and BME. The clonal expansion of mast cells in the bone marrow leads to a strong inflammatory milieu in the bone marrow with osteoclast activation, but also inflammation-based bone pain. ISM is characterized by a typical rash (urticaria pigmentosa) and/or a history of anaphylaxis, as well as increased serum tryptase. However, it can only be diagnosed by bone marrow biopsy. Cytotoxic treatment of ISM is rarely needed and bone disease is usually treated with bisphosphonates.

### 6.6. Ischemic/Metabolic BME

Ischemic BME comprises avascular osteonecrosis and (Charcot) neuro-osteoarthropathy. The diagnosis of avascular necrosis (AVN) can be challenging and relies on MRT and CT imaging. AVN can be classified according to the ARCO criteria [67]. Although BME is often thought to be an early sign of AVN, recent studies showed that BME occurs only in late stages of the disease, i.e., ARCO stages III and IV, as a sign of biomechanical deterioration of the trabecula [12,13,14,15]. AVN can occur in the absence of trauma or following a fracture with disruption of the vascular supply. The exact pathophysiology of atraumatic AVN is unclear, but numerous risk factors have been identified, including glucocorticoid use, alcohol consumption, trauma, chemotherapy, kidney transplants and coagulation abnormalities, including sickle cell diseas [67]. Treatment of AVN depends on the joint affected, the ARCO stage and the size of the affected region within the bone. Overall, joint preserving procedures are only indicated in early stages (and localized AVN) and include physiotherapy, off-loading, medication, and various types of surgery. For advanced stages of AVN with severe joint destruction, joint replacement or arthrodesis, depending on the joint affected, are recommended [68,69,70]. Charcot neuro-osteoarthropathy affects the foot and ankle and is a devastating consequence of peripheral polyneuropathy. It manifests as an aseptic inflammation and progressive degeneration of the bone. In its active stages, patients present with a pain-free red and swollen foot and with marked BME. If not diagnosed early and treated properly, the impaired bone quality results in spontaneous fatigue bone fractures with subsequent deformation and considerable reduction in life expectancy [71,72]. The pathogenesis is largely unknown, but an unphysiological osteoclast to osteoblast ratio and an increased immuno-reactivity (IL1, IL6, TNF) have been shown [72]. Recent studies have focused on the RANK/RANK-L/OPG system [73,74]. The current treatment recommendations comprise extended immobilization and avoidance of weight-bearing activities in the affected leg for a duration of 3–12 months [72]. Following bony consolidation, the primary long-term aim is to prevent ulceration.

### 6.7. Metabolic BME

If no cause for the painful BME could be found during the preceding diagnostic evaluation, or if it shows a refractory, relapsing and or migratory course, a metabolic (primary/secondary osteoporosis) must be ruled out. *Vitamin D deficiency* is very common, especially in the elderly population. In contrast, secondary hyperparathyreoidism with severe osteolomalacia and typical pseudofractures (Looser’s zones) is rather rare in developed countries. Current guidelines recommend avoiding even moderate and mild vitamin D deficiency in patients at risk of (micro-) fractures. A similar condition of impaired bone mineralization is adult hypophosphatasia (aHPP). The clinical appearance in adults can include musculoskeletal pain and (fragility-)fractures, most frequently metatarsal, femur and stress fractures [75]. About 14% suffer from dental abnormalities [76]. In patients with low alkaline phosphatase (ALPL) levels, determination of pyridoxal-5′-phosphate (and/or phosphoethanolamine) levels, which can be reduced by a genetic testing of a possible ALPL gene mutation, is recommended. In aHPP, low (further osteomalacia) and also high (risk of hypercalcemia and/or extraosseous calcification) vitamin D levels should be avoided. Bisphosphonates inhibit alkaline phosphatase and should also be avoided [77]. In very severe cases, with first signs already in childhood, a specific treatment with asfotase alpha is possible. Primary hyperparathyroidism (pHPT) presents with the well-known clinical triade “bones, stones and abdominal groans”. BME is detectable in a significant proportion of patients with asymptomatic primary hyperparathyroidism [78]. Laboratory workup reveals high serum calcium accompanied with low phosphorus and high alkaline phosphatase and intact parathyroid hormone (iPTH) levels. PHPT may be surgically cured by the removal of the parathyroid adenoma. With respect to the bone, secondary hyperparathyroidism causes similar changes, due to the high iPTH activity. Vitamin D deficiency, as one major cause, is discussed above. The second major cause is renal insufficiency, which ultimately causes a complex bone disorder called chronic kidney disease–mineral and bone disorder (CKD-MBD). Although data regarding the incidence of BME in the several CKD stages are scarce, CKD-MBD may accompany BME, as suggested by animal models [79]. In addition, CKD-MBD causes an osteoporosis-like syndrome with impaired bone quality and increased risk of fragility (micro-) fractures. CKD-MBD is treated according to the international KDIGO guideline [80]. High cortisol levels, either as iatrogenic (medication) or as endogenous hypercortisolism (Cushing’s disease, ectopic or adrenal), cause severe impairment of bone metabolism with increased risk of fractures and avascular necrosis of the bone. Cushing’s syndrome includes clinical signs like easy bruising, facial plethora, proximal myopathy and striae rubrae, in addition to the less specific signs like obesity, glucose intolerance, osteoporosis and hypertension. In the case of clinical suspicion, a 1 mg overnight dexamethasone suppression test (or late night salivary cortisol or 24 h urin free cortisol) is recommended [81]. Not only glucocorticoids, but also other drugs, have been described to be associated with the incidence of BME, including busulfan, calicineurin inhibitors, 13-cis-retinoid acid, everolimus, hydroxyurea, imatinib and interferon [82,83,84].

Some of the above mentioned secondary BME causes are also possible causes of secondary osteoporosis. Osteoporosis is defined as increased risk for (micro-) fracture which in turn may cause secondary BME. Cross-sectional studies suggest that prevalent general osteoporosis is a substantial risk factor for BME. Terms such as transient, migrating or local osteoporosis, should be avoided, as they have been inconsistently used for metabolic causes and BMES. Of note, diagnostic sensitivity of DXA is low with more than 50% of those patients suffering a typical fragility fracture haveing normal or only osteopenic DXA results [85]. Postmenopausal hypogonadism and age are the most frequent causes of osteoporosis, but there are many other medical conditions that increase fracture risk, such as prevalent osteoporotic fracture, parental history of hip fracture, low BMI, smoking, alcohol, rheumatoid arthritis, diabetes mellitus and medications like glucocorticoids, aromatase inhibitors or antiandrogens. Although quite rare, pregnancy-associated osteoporosis plays an important role for BME differential diagnosis. Pregnant women may develop BME (“pregnancy-associated transient osteoporosis of the hip”) in the third trimester or immediately postpartum, especially in the pelvic bone [86]. Pregnancy-associated osteoporosis is a severe disease, typically occurring during the first pregnancy, sometimes accompanied by multiple fractures [87].

## 7. BME in the Pediatric Population

### 7.1. Non-Pathogenic BME in Children

In healthy children, BME is mostly a transient, self-limiting condition. Focal peripheral edema at the growth plate, particularly observed in the knee, is attributed to early stages of growth plate closure, and may be associated with pain. However, this phenomenon should not be regarded as pathogenic and does not require invasive diagnostic procedures or imaging follow-up. Similarly, local BME can also be found in children during rapid growth at in the spine (most often cervical) [88], the epi- and metaphysis of the distal femur, proximal tibia, or proximal fibula and extending into both [89], and at carpal bones/wrists [90,91] as a transient finding, not related to physical activity, and decreasing over time.

Potential pathomechanisms include formation of metaplastic bone-neofibrocartilage, diminution in bone flexibility [89], local vascular disturbance, microtrauma, or bone contusion [92] and age-related response to biomechanics [90].

### 7.2. Pathogenic BME in Children

Most underlying conditions for BME in children overlap with those already mentioned above for adults. Briefly, the underlying diseases described in the literature and/or typical for pediatric population are: Primary inflammatory/rheumatological: CRMO juvenile idiopathic arthritis [93]. Metabolic disease: Gaucher disease. Bone disease: osteofibrous disease, osteoid osteoma [94,95,96,97,98]. Neoplastic: Langerhans cell histiocytoma, leukemia-associated osteoarthralgia, sarcoma [99,100]. Osteonecrosis: 15–47% of pediatric patients after high-dose glucocorticoids. BME might be a marker of potential progression for advanced osteonecrosis [101] and is associated with subchondral fractures. It is therefore of use to stratify patients to joint-preserving interventions: absence could justify a wait and watch approach [102].

## 8. Bone Marrow Edema Syndrome (BMES)

(BMES, migrating BMES, transient BMES, transitory BMES, transient osteoporosis or primary BME are interchangeable terms for the same entity of BME [48,103,104,105,106,107]. All refer to a temporary (transient) painful BME, without any evidence of focal osteonecrosis or a specific underlying pathology [48,106]. Therefore, the LMU Consensus Group recommends that the term BMES be used only for a painful BME without an identifiable underlying cause according our diagnostic algorithm. Hence, BMES is a diagnosis of exclusion.

BMES is a rare condition and, consistent with its definition, the etiology remains largely unknown. It is mainly reported in the lower extremity of middle-aged males [2,8]. The most common sites, in descending order, are hip, knee, ankle, and foot [108]. It can appear only at a single bone or as a multifocal BME. In rare cases, BMES can be “migratory,” i.e., showing multiple episodes in different locations [3,10,109].

The primary aim of treating BMES is a quick reduction of pain and disability, as well as dissolution of the BME [106]. Usually, BMES is considered self-limiting over 3 to 18 months [24,106]. Measures to shorten this natural history should initially be partial weight-bearing, immobilization, analgesics and anti-inflammatory medication. Additional treatment approaches include extracorporeal shock wave therapy, bisphosphonates, and iloprost [1,3,29,110,111,112]. Regarding pharmaceutical therapy, we recommend zoledronate 5 mg IV as primary treatment and iloprost administered for a total dose of 180 μg IV over 5 days (day 1: 20 μg over 6–8 h; day 2–5: dose increase to 40 μg per day over 6–8 h). Surgical core decompression should be considered only after failure of the above-mentioned treatments in refractory cases [113,114].

These pharmaceutical therapies can potentially also be applied in BME with a known underlying cause, in a shared decision between the patient and the treating physician, e.g., when the standard treatments are not effective, or the therapy is highly invasive with the patient requesting non-surgical treatment.

## 9. Conclusions

To identify the underlying cause of painful BME is the paramount prerequisite to initiating the appropriate treatment. As the underlying diseases encompass almost all medical specialities, patient journies and initiation of the correct treatment are often considerably prolonged. The LMU Consensus Group, consisting of nine medical specialities, compiled, for the first time, an evidence-based diagnostic algorithm for painful BME. This step-wise approach aims at reducing the number of unneccissary diagnostics and at accelerating the time to correct diagnosis and treatment initiation. This single-center perspective should not be considered the current gold standard, but rather serve as a basis for further discussion.

## Figures and Tables

**Figure 1 jcm-09-00551-f001:**
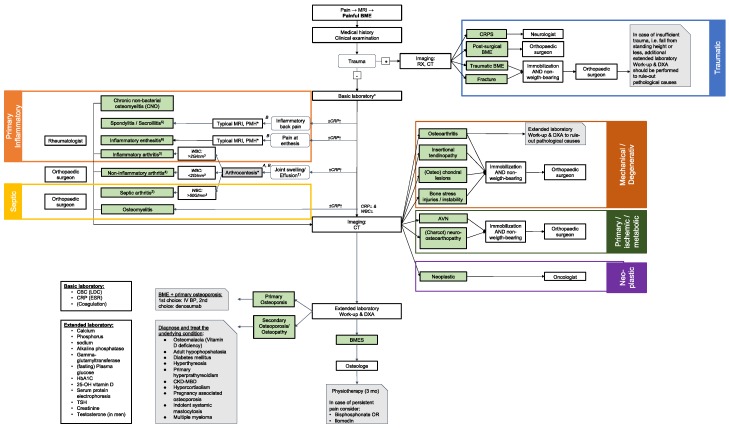
LMU consensus algorithm for the diagnosis and management of BME. Accessible to arthrocentesis; (1) Any swollen joint should be assessed for effusion and subsequently an arthrocentesis performed; (2) Typically associated with WBC counts >50 G/mm3, (or <20 G/mm3) but lower counts possible; (3) e.g., rheumatoid arthritis, psoriatic arthritis, gout; (4) e.g., osteoarthritis; (5) Diagnosis of spondyloarthritis based on ASAS criteria; (6) Either in the context of primary inflammatory disease, e.g., psoriatic arthritis, SpA or stress-induced; A. council orthopedic surgeon; B. council rheumatologist; Abbreviations: AVN: avascular necrosis; BME: bone marrow edema; CRP: C-reactive protein; CRPS: complex regional pain syndrome; DXA: dual-energy X-ray absorptiometry; CT: computer tomography; Diff. blood count: differential blood count; ESR: erythrocyte sedimentation rate; Mech/Deg: mechanical/degenerative; MRI: magnetic resonance imaging; WBC: white blood cell.

**Table 1 jcm-09-00551-t001:** Ludwig Maximilians University LMU Consensus classification of BME according to their etiology.

Category (Section)	Etiology [25]
Traumatic (Section 6.1)	Traumatic BME or (micro-) fracture BME with/without osteoporosis
	Post-surgical BME
	Complex regional pain syndrome (CRPS)
Septic (Section 6.2)	Osteomyelitis
	Septic arthritis
Primary inflammatory (Section 6.3)	(Peripheral) arthritis
	Spondylitis/sacroiliitis
	Enthesitis
	Chronic non-bacterial osteomyelitis (CNO)
Mechanical/Degenerative (Section 6.4)	Osteoarthritis
	Insertional tendinopathy
	(Osteo)chondral lesions
	Bone stress injuries/Instability
Neoplastic (Section 6.5)	Primary or secondary benign or malignant bone tumors
Ischemic/Neurogenic (Section 6.6)	Avascular osteonecrosis
	(Charcot) Neuro-osteoarthropathy
Metabolic (Section 6.7)	Primary osteoporosis
	Secondary osteoporosis/osteopathy
Diagnosis by exclusion (Section 8)	Bone marrow edema syndrome

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
