# Peer review of "How We Manage Bone Marrow Edema—An Interdisciplinary Approach"

_jcm, 2020, doi:10.3390/jcm9020551_

Round 1
Reviewer 1 Report
GENERAL COMMENTS
It is commendable that this group has joined together to address the clincial issues surrounding the detection of Bone marrow Lesions. The authors are outlining the huge number of clinical situations where BMLs may be detectable and highlight conditions where acute intervention is warranted. The review is detailed, but the balance between descriptions can be discussed. For example I find the space alotted to osteoporosis extensive. The description of causes of secondary osteoporosis is excessive, a simple table would suffice as BMLs in osteoporosis are virtually all fracture related. The authors promote a conservative approach despite several papers showing clear shortening of pain duration in f.ex. transient osteoporosis using bisphosphonates (ref.20) and reduction of BML size in OA (Varenna, M., et al. (2015). Rheumatology (Oxford) 54(10): 1826-1832). Therefore I prefer iv. zoledronic acid over Iloprost), but there is a lack of clinically controled studies on BMLS treatment.
SPECIFIC COMMENTS
I think it should be mentioned that BMLs are also visible using Ultrasound I find that a more detailed description of the histology of BMLs is warranted. I would recommend to included the newer histology studies (e.g. Shabestari, M., et al. (2016). Osteoarthritis Cartilage 24(10): 1745-1752)
Author Response
Thank you very much for the insightful review of our manuscript and the favorable uptake of the paper. We have carefully assessed the comments of the reviewers and revised the manuscript accordingly. A detailed point-by-point response to the individual comments is outlined below.
Point-by-point reply to the Reviewers' comments:
Similar remarks from Reviewer #1 and Reviewer #2:
“English language and style are fine/minor spell check required”
Thank you very much for this comment, and please excuse language mistakes. As we are all non-natives, English language and style was checked by a native speaker.
Reviewer #1 “…I find the space allotted to osteoporosis extensive. The description of causes of secondary osteoporosis is excessive, a simple table would suffice as BMLs in osteoporosis are virtually all fracture related.”
Reviewer #2 “secondary causes
This section is too long, really boring and doesn't help the reader to solve the problem this feels a no ending list. i think authors should focus on 2-3 situation evaluating treatment and every steps and not report every cases.”
We thank both reviewers for their similar feedback on the extent of the “secondary causes” section. As a result, we have substantially shortened this section to be more concise and improve readability of the manuscript. Notwithstanding, a central concern of the consensus group was the identification of secondary/ -potentially treatable causes of BME along with evidence-based treatment options. Another important conclusion was that a single medical discipline managing the individual case is usually not able to cover all secondary BME causes. Therefore, based on a literature review and several expert discussions, we elucidated a diagnostic flow-chart and a list of clinical conditions with clear BME association and described them. Since the descriptions of secondary causes represent the backbone of this work (also the diagnostic/ therapeutic algorithm), we would advocate to not reduce this important part too much, in order to preserve the evidence-based aspect of this article. However, we substantially shortened this section.
We agree with Reviewer 1 that many of the mentioned secondary BME causes also represent secondary causes of osteoporosis, and (micro-) fractures may be an important mechanism of BME in these cases. However, in many instances the suggested pathophysiology is different and more probably relies on local inflammation. Therefore, we picked up this valuable comment and separated the osteoporosis part and moved it to the end of this section.
_____
Reviewer #1 comments:
“The authors promote a conservative approach despite several papers showing clear shortening of pain duration in f.ex. transient osteoporosis using bisphosphonates (ref.20) and reduction of BML size in OA (Varenna, M., et al. (2015). Rheumatology (Oxford) 54(10): 1826-1832). Therefore, I prefer iv. zoledronic acid over Iloprost), but there is a lack of clinically controlled studies on BMLS treatment.”
Thank you very much for this comment. We totally agree with the reviewer, that bisphosphonates (i.e. i.v. zoledronic acid) should be considered first-line medical treatment. We do state this in line 422: “Regarding pharmaceutical therapy, we recommend zoledronate 5mg IV as primary treatment”. Still, as bisphosphonates as well as iloprost, are used off-label, and BMES has to be considered a self-limiting disease in general, we must recommend non-medical treatment, i.e. “partial weight-bearing, immobilization, analgesics and anti-inflammatory medication. […] extracorporeal shock wave therapy”, as a general first line treatment approach.
With respect to the BML observed in OA, we added the references and further elaborated on the positive effects of bisphosphonates on BML reduction and pain. Still, long-term data on the influence of bisphosphonates on the progression of OA, i.e. the time to total knee replacement, are missing. Therefore, we do not believe, bisphosphonates, as an off-label therapy for OA (missing a diagnosis of osteoporosis), can be recommended as the current gold standard.
“I think it should be mentioned that BMLs are also visible using Ultrasound I find that a more detailed description of the histology of BMLs is warranted. I would recommend to include the newer histology studies (e.g. Shabestari, M., et al. (2016). Osteoarthritis Cartilage 24(10): 1745-1752)”
We thank the reviewer for this suggestion and included a statement regarding the use of ultrasound in the detection of BME in the manuscript. Further, newer histology studies were included as suggested.
In summary, we have addressed the concerns. We believe that the revisions have clarified the issues raised by the reviewer, none of which challenged the importance or validity of the paper. Indeed, we believe that addressing the comments of the reviewers and implementing new data have helped to make the presentation of the data more precise. This most likely further increased the interest of our paper to a broad, multidisciplinary readership.
Therefore, we hope that the revised manuscript is now acceptable for publication in Journal of Clinical Medicine. In case any further questions remain, please feel free to contact us at any time!
Reviewer 2 Report
Abstract:
the abstract is not clear. is the manuscript a review? an expert opinion? what the authors want to demonstrate?
Introduction:
in lower extremity please report the following reference ( d'ambrosi et al. the role of bone marrow edema on osteochondral lesions of the talus)
introduction is confusing. authors focus on ethiopatogensis than on the treatment without specifying. there is no connection between paragraphs
histhopathology
too much repetitions. this section can be shortened without loosing informations
imaging models
please help the readers to understand which is the better way to study BME and not reporting a list of all types of imaging
classification
this section is really interesting and well done
diagnostic steps
ok
secondary causes
this section is too long, really boring and doesn't help the reader to solve the problem this feels a no ending list. i think authors should focus on 2-3 situation evaluating treatment and every steps and not report every cases
conclusions are missing
Author Response
Thank you very much for the insightful review of our manuscript and the favorable uptake of the paper. We have carefully assessed the comments of the reviewers and revised the manuscript accordingly. A detailed point-by-point response to the individual comments is outlined below.
Point-by-point reply to the Reviewers' comments:
Similar remarks from Reviewer #1 and Reviewer #2:
English language and style are fine/minor spell check required”
Thank you very much for this comment, and please excuse language mistakes. As we are all non-natives, English language and style was checked by a native speaker.
Reviewer #1 “…I find the space allotted to osteoporosis extensive. The description of causes of secondary osteoporosis is excessive, a simple table would suffice as BMLs in osteoporosis are virtually all fracture related.”
Reviewer #2 “secondary causes
This section is too long, really boring and doesn't help the reader to solve the problem this feels a no ending list. i think authors should focus on 2-3 situation evaluating treatment and every steps and not report every cases.”
We thank both reviewers for their similar feedback on the extent of the “secondary causes” section. As a result, we have substantially shortened this section to be more concise and improve readability of the manuscript. Notwithstanding, a central concern of the consensus group was the identification of secondary/ -potentially treatable causes of BME along with evidence-based treatment options. Another important conclusion was that a single medical discipline managing the individual case is usually not able to cover all secondary BME causes. Therefore, based on a literature review and several expert discussions, we elucidated a diagnostic flow-chart and a list of clinical conditions with clear BME association and described them. Since the descriptions of secondary causes represent the backbone of this work (also the diagnostic/ therapeutic algorithm), we would advocate to not reduce this important part too much, in order to preserve the evidence-based aspect of this article. However, we substantially shortened this section.
We agree with Reviewer 1 that many of the mentioned secondary BME causes also represent secondary causes of osteoporosis, and (micro-) fractures may be an important mechanism of BME in these cases. However, in many instances the suggested pathophysiology is different and more probably relies on local inflammation. Therefore, we picked up this valuable comment and separated the osteoporosis part and moved it to the end of this section.
Reviewer #2 comments:
“Abstract: The abstract is not clear. is the manuscript a review? an expert opinion? what the authors want to demonstrate?”
We appreciate your request for clarification. Consequently, we stated in the abstract that our consensus paper on the diagnosis and management of BME finally represents a review. We hope the changes allow the reader to comprehend the nature of our paper.
“Introduction: In lower extremity please report the following reference (d'ambrosi et al. the role of bone marrow edema on osteochondral lesions of the talus)”
We thank the reviewer for this suggestion and implemented this important publication. The reference was added to the appropriate section ((Osteo)chondral lesions)
“Introduction is confusing. authors focus on ethiopatogensis than on the treatment without specifying. there is no connection between paragraphs”
Thank you very much for this comment. The manuscript was proofread and the introduction reworked to provide a more cohesive background for the manuscript.
“Histhopathology too much repetitions. this section can be shortened without losing informations”
We agree that this section could be more concise. Therefore, we deleted repetitive parts. Further, we added a new additional Supplemental table (Suppl. Table 2) showing the results of a comprehensive literature search on the (histo)pathology of BME per the different etiologies.
“Imaging models: Please help the readers to understand which is the better way to study BME and not reporting a list of all types of imaging”
The “Imaging modalities” sections aims at giving readers with limited experience in radiographic modalities a broad overview. Per your valuable comment, this section was shortened. In the following, when outlining the proposed, step-wise diagnostic algorithm, the necessary imaging modalities are outlined. The aim of this step-wise approach is to reduce the number of unnecessary diagnostics and to systematically guide readers to the right diagnosis.
“Classification: This section is really interesting and well done”
“diagnostic steps ok”
We thank the reviewer for the appreciation of our manuscript.
“Conclusions are missing”
We thank the reviewer for this remark. Please excuse this short-coming. We added a conclusion at the end of the manuscript.
In summary, we have addressed the concerns. We believe that the revisions have clarified the issues raised by the reviewer, none of which challenged the importance or validity of the paper. Indeed, we believe that addressing the comments of the reviewers and implementing new data have helped to make the presentation of the data more precise. This most likely further increased the interest of our paper to a broad, multidisciplinary readership.
Therefore, we hope that the revised manuscript is now acceptable for publication in Journal of Clinical Medicine. In case any further questions remain, please feel free to contact us at any time!
Round 2
Reviewer 2 Report
authors answered in full to all reviewers queries